# Census Demographics and Chlorpyrifos Use in California’s Central Valley, 2011–15: A Distributional Environmental Justice Analysis

**DOI:** 10.3390/ijerph17072593

**Published:** 2020-04-10

**Authors:** Daniel J. Hicks

**Affiliations:** Data Science Initiative, University of California, Davis, CA 95616, USA; hicks.daniel.j@gmail.com

**Keywords:** pesticides, chlorpyrifos, environmental justice, California, spatial regression, public data

## Abstract

Chlorpyrifos, an acetylcholinesterase inhibitor (ACI), is one of the most widely used insecticides in the world, and is generally recognized to be a moderate human neurotoxin. This paper reports a distributional environmental justice (dEJ) analysis of chlorpyrifos use in California’s Central Valley, examining the way distributions of environmental risks are associated with race, ethnicity, class, gender, and other systems of structural oppression. Spatial data on chlorpyrifos use were retrieved from California’s Department of Pesticide Registration public pesticide use records for 2011–2015. These data were combined with demographic data for the Central Valley from the American Community Survey (ACS). Spatial regression models were used to estimate effects of demographic covariates on local chlorpyrifos use. A novel bootstrap method was used to account for measurement error in the ACS estimates. This study finds consistent evidence that Hispanic population proportion is associated with increased local chlorpyrifos use. A 10-point increase in Hispanic proportion is associated with an estimated 1.05–1.4-fold increase in local chlorpyrifos use across Census tract models. By contrast, effects of agricultural employment and poverty on local chlorpyrifos use are ambiguous and inconsistent between Census tracts and Census-designated places.

## 1. Introduction

Chlorpyrifos, an acetylcholinesterase inhibitor (ACI), is one of the most widely used insecticides in the world. In California in 2016, it was the 29th most heavily used pesticide active ingredient, with over 900,000 pounds applied over 640,000 acres [1]. Like several other organophosphate (OP) pesticides, it is generally recognized to be a moderate neurotoxin (p. 67, [2,3,4,5]), (and further citations in, [6]). Bellinger [7] estimates an expected loss of 4.25 IQ points in children for each order-of-magnitude increase in maternal urinary concentration of dialkyl phosphate (DAP) metabolites from OP pesticides. Chlorpyrifos was banned from residential use in the US in 2001.

Because of this evidence of harm and continued widespread use, chlorpyrifos is a significant topic of regulatory controversy. In 2007 the environmental organizations Pesticide Action Network North America (PANNA) and Natural Resources Defense Council (NRDC) filed a petition with US EPA, calling on the agency to revoke all tolerances for chlorpyrifos, effectively banning it. In 2017, US EPA rejected this petition [8]. In 2018, Hawai’i and California both proposed state-level restrictions on use of the chemical active substance. Hawai’i’s complete ban comes into effect in 2023, with greater restrictions beginning in 2019 [9]. California proposed classifying chlorpyrifos as a toxic air contaminant and prohibiting aerial applications [10], though as of March 2019 these proposed restrictions have not been adopted.

This paper reports an distributional environmental justice (dEJ) analysis of chlorpyrifos use in California’s Central Valley. A dEJ analysis examines the way the distribution of environmental risks intersect with race, ethnicity, class, gender, and other systems of structural oppression. Since the landmark report “Toxic Wastes and Race in the United States” [11], a significant dEJ scholarly literature has emerged, documenting numerous inequitable distributions of multiple forms of environmental hazards [12,13,14,15,16,17]. Specifically, this study asked to what degree community demographic characteristics—including but not limited to race, ethnicity, class, gender, and age—are associated with increased (or decreased) potential exposure to chlorpyrifos.

This study used spatial regression techniques to examine the distribution of chlorpyrifos use across California’s Central Valley. While spatial methods are frequently used in dEJ analysis, they are often not statistically sophisticated [15,18], making them vulnerable to technical criticism [19,20], which can limit their effectiveness as tools for policy change or legal remediation.

It is important to recognize that environmental justice issues are not exhausted by the distribution of environmental hazards. Schlosberg [21], drawing on previous work by Young [22] and Shrader-Frechette [13], argues that environmental justice also includes procedural justice and appropriate recognition and respect for community identity. For example, racialized communities that are outside of an administrative district—and so formally excluded from land-use decisions within the district—might be exposed to pollution emitted as a result of those land-use decisions [23]; this is a form of procedural injustice. Or, communities’ claims and arguments might be ignored because they are racialized or lack formal scientific credentials [24]. This is an example of misrecognition and disrespect.

However, dEJ remains an important aspect of EJ, and the kinds of quantitative methods deployed in this project can be especially useful for identifying distributive environmental injustices.

Previous spatial analyses of chlorpyrifos use and exposure in California fall into two categories. First, physical–chemical simulation methods have been used to develop fate-and-transport estimates of chlorpyrifos presence across the entire state. For example, Luo and Zhang [25] used public chlorpyrifos use data and a fate-and-transport simulation to estimate how the chemical moves through space. However, this study did not examine the population exposed to the pesticide, and therefore was not a dEJ analysis. In contrast, the current study focuses on demographic covariates for chlorpyrifos use in the Central Valley, and thus is more focused on the population that is potentially exposed than on the chemical itself.

The other category of studies use comparatively small-scale epidemiological methods to examine the public health impacts of chlorpyrifos exposure. Several studies in this category have been conducted as part of the Center for the Health Assessment of Mothers and Children of Salinas (CHAMACOS) Study, based at University of California, Berkeley. Over the past 20 years, the CHAMACOS Study has followed roughly 800 children in a farmworker community in California’s Salinas Valley, a major agricultural region south of the San Francisco Bay Area and west of the Central Valley [26]. (The Salinas Valley is located primarily in Monterey County; Monterey County is one of the state’s most agriculturally productive counties, notably for producing strawberries, lettuce, broccoli, and cauliflower (p. 5, California Department of Food and Agriculture [27].)) Gunier et al. [28] compared public pesticide use records to Wechsler Intelligence Scale for Children (WISC) scores for 255 7-year-old children. Examining a 1 km buffer around the residence of pregnant women participants, they found that a 1 standard deviation increase in OP use (including chlorpyrifos) in this buffer during pregnancy was associated with a 1–4 point decrease in WISC scores (see also [29]).

Because the CHAMACOS study focuses on populations that are likely to be highly exposed or socially vulnerable to chlorpyrifos impacts, it can be considered a dEJ study. In publications to date, the CHAMACOS study focused on estimating the human health impacts of chlorpyrifos exposure, rather than relative or absolute degree of exposure. In contrast, the current study considers social vulnerability as a predictor for potential chlorpyrifos exposure. The current study also works at a much larger scale, analyzing data for more than a thousand Census tracts and more than a million uses of chlorpyrifos across more than 10,000 square miles.

Methodologically, the current study closely resembles a number of other studies that have used spatial methods to identify demographic predictors of potential exposure to other kinds of environmental health hazards (and benefits), including toxic releases [30,31]; air pollution [32,33,34] and noise pollution [35]; hydraulic fracturing wells [36]; water use restrictions [37] and cumulative pollution burden [38]; as well as access to environmental amenities such as parks [39,40]. Often these studies have been framed explicitly in terms of environmental justice. For example, Liévanos [41] used data from across the continental US and spatial methods to identify clusters of high lifetime cancer risk (LCR) due to air pollution, then (non-spatially) regressed these clusters against composite Census tract demographic variables. Liévanos [41] concluded that “isolated Latino immigrant-economic deprivation is the strongest positive demographic predictor of tract presence in air-toxic LCR clusters, followed by black-economic deprivation and isolated Asian/Pacific Islander immigrant-economic deprivation” (p. 50, [41]), a significant dEJ finding.

All together, I have not been able to identify any prior studies that apply spatial methods to examine environmental justice aspects of pesticide use and potential exposure.

## 2. Materials and Methods

The primary analysis of this study was a spatial regression of potential chlorpyrifos exposure against Census demographic data. The software language R (3.6.1) was used to clean and analyze all data, with especially notable use of the tidyverse (1.2.1), tidycensus (0.9.6), sf (0.8-1), spdep (1.1-3), and tmap (2.3-1) packages [42,43,44,45,46]. Complete cleaning and analysis code is available at https://github.com/dhicks/chlorpyrifos/releases/tag/v1.1.

### 2.1. Study Area

The study area for this project was California’s Central Valley. California is a major US agricultural producer, producing over 13% of US agricultural value (p. 2, [27]). The Central Valley is the largest center of California’s agricultural production, containing 7 of the state’s 10 most agriculturally productive counties (p. 5, [27]). Consequently, the Central Valley is also a major user of pesticides, including chlorpyrifos. In 2013–15, on average 1.3 million pounds of chlorpyrifos active ingredient was used in California annually, 1.1 million pounds of which was used in the Central Valley (Table 6, [47]). By crop, the heaviest uses of chlorpyrifos during this period were on almond, alfalfa, walnut, orange, and cotton (Table 4, [47]). Demographically, the Central Valley is home to substantial populations of both Hispanic and non-Hispanic White residents, which raises the possibility of inequitable distributions of pesticide exposure, i.e., distributive environmental injustice. In addition, California’s Department of Pesticide Regulation (DPR) makes available public, detailed, geocoded data on pesticide use in the state. Combined with public data from the US Census, this makes it straightforward to retrieve data for a pesticide-related dEJ study.

The Central Valley can be defined in a number of different ways. Since the units of analysis for this study are tracts and places designated by the US Census, a county-based definition was judged to be most appropriate. Sacramento County was excluded because, unlike the rest of the region, most of its area is urban. Seventeen other counties were used to define the Central Valley; see Table 1 and Figure 1.

### 2.2. Pesticide Use

In California, chlorpyrifos is banned for residential use, and more than 99% of chlorpyrifos use is in commodity agriculture (Table 2, Segawa and Wofford [47]). California’s Department of Pesticide Regulation (DPR) releases annual public datasets for agricultural pesticide use across the state, known as Pesticide Use Reports (PUR) (http://www.cdpr.ca.gov/docs/pur/purmain.htm). These data exclusively report agriculture use; this limitation was judged to be acceptable for studying chlorpyrifos.

Full datasets for 2011–2015 were retrieved, combined, and filtered on chemical name for chlorpyrifos. To avoid edge effects, PUR data for all counties across the state were used, not just the Central Valley. For example, while Sacramento County was not included in the study area, there is some chlorpyrifos use in the southeastern part of Sacramento County, very close to areas of San Joaquin County that are included in the study area. These Sacramento County uses were incorporated into the analysis.

In the PUR data, chlorpyrifos uses are spatially linked to townships and sections; annual and all-study-period active ingredient totals at the centroid of each 1 mile-square (1.6 km × 1.6 km) section were calculated. Because these centroids do not match the actual use locations (i.e., farm fields), the centroid totals might be unreliable for the smallest CTD, 1 km (see discussion and Table 2 below). However, this error is expected to be negligible for the other CTDs.

All together, 1,113,398 use records for chlorpyrifos were identified in the DPR datasets for 2011–2015. After aggregating by sections and years, there were 31,789 records, with annualized use values ranging from $10^{-2}$ to $10^{4}$ lbs of active ingredient.

### 2.3. Demographics

The second primary dataset comprised American Community Survey (ACS) five-year estimates, from 2011–2015, for all Census tracts and places in the 17 Central Valley counties.

For each tract and place, estimates and margin of error (MOE) values were retrieved for four categories of demographic variables: *race and ethnicity* (Hispanic, non-Hispanic White, non-Hispanic Black, Indigenous, and Asian residents), *foreign-born noncitizens*, *children under 5* (who may be especially sensitive to chlorpyrifos exposure due to small body weight and critical neurodevelopmental stages), and *poverty* (individuals with an Census-determined income/poverty ratio below 1). Because PUR data come only from agricultural uses, agricultural employment was also retrieved as a potential control. (The selection of independent variables and controls is discussed in Section 2.6).

Population densities and proportions (e.g., the fraction of all residents who are Hispanic) were calculated for the total population and each of these ACS variables for every tract and place, using Census-recommended methods to calculate MOEs for these derived variables (p. 11ff, [48]). Nine tracts and 15 places with estimated total population or total employment of 0 were excluded; 391 places and 1044 tracts were used in all further analysis.

Because much of the study area is rural, tract size and population density vary over four orders of magnitude, from 0.8 to 5600 residents per ${\mathrm{km}}^{2}$. Places cover 87% of the population, including 83% of non-Hispanic White and 88% of Hispanic residents, with population densities between 1.4 and 4400 residents per ${\mathrm{km}}^{2}$. However, places are geographically separated from each other, covering only 8% of the area of the tracts; in constructing contiguity-based spatial weights, 62% of places had no neighbors. To mitigate the tradeoff between coverage and accuracy, both tracts and places were used in parallel analyses in the remainder of this study.

The Modifiable Areal Unit Problem (MAUP) has been used to criticize dEJ projects (p. 41ff, [17,20]). “Egocentric neighborhood” methods have been used to address the MAUP in segregation research  [49]. However, these methods assume that populations are distributed evenly within each discrete region (e.g., each Census tract). This assumption is inappropriate for this project, which includes many spatially heterogeneous rural regions. In addition, the large geographic scale of this project would require millions of egocentric neighborhoods, and so trillions of distance calculations between neighborhoods and chlorpyrifos uses. The tract-level distance calculations already pressed the limits of the computing power available for this phase of the project. More fine-grained regions (e.g., Census block groups or blocks) would have multiplied uncertainty in the ACS estimates, and also likely would have exceeded the available computing power.

As a compromise, block population counts from the 2010 Census were used to calculate weighted centroids for each tract and place. These centroids more accurately represent the “average location” of the population in each tract, without requiring more computing power in the distance calculation step.

Chlorpyrifos use section centroids, Census tracts, and places included in the study area are shown in Figure 1.

### 2.4. Linking Pesticide Use to Tracts and Places

Chlorpyrifos use was linked to tracts and places using the concept of *characteristic travel distance* (CTD). CTD is defined as the distance “at which 63% of the original mass of [a] volatilized [chemical] is degraded or deposited” [50]. In the simplest sense, CTD models assume exponential decay:(1)$$\begin{array}{cc} q(d) & =q_{0}\beta^{d} \end{array}$$
where q(d) is the quantity of a chemical at a distance *d* from its source; $q_{0}$ is the quantity volatilized at the source; and β is a constant related to the CTD:(2)$$\begin{array}{cc} \left(1-0.63\right) & =\beta^{CTD} \end{array}$$(3)$$\begin{array}{cc} \beta & =0.37^{1/CTD} \end{array}$$

$\beta^{d}$ is referred to here as the *decay coefficient*. Given a CTD, the decay coefficient can be used in Equation (Equation 1) to calculate the *distance-weighted local use* of chlorpyrifos at a location (tract or place centroid) *i*:(4)$$\begin{array}{cc} q_{i} & =\sum_{u}q_{u}\beta^{d_{iu}} \end{array}$$(5)$$\begin{array}{cc} \; & =\sum_{u}q_{u}0.37^{d_{iu}/CTD} \end{array}$$
where $q_{u}$ is the total use at section *u* and $d_{iu}$ is the Euclidean distance between the centroid of *i* and the centroid of *u*. To slightly account for the fact that the residents of a location are not located at its centroid, the decay coefficient is set to 1 whenever the centroid associated with use *u* is within *i*, regardless of $d_{iu}$.

It is important to stress that Equations (Equation 1) and (Equation 4) provide, at best, rough estimates of potential exposure, that is, that amount of chlorpyrifos that residents of a tract or place *might* have been exposed to. These estimates do not take into account prevailing or occurrent winds, or other chemical transport media such as water. Diverse and variable processes of application, fixation, and chemical transformation are represented as simple exponential decay. By comparison, Luo and Zhang [25] use a physical–chemical model and PUR data to produce more sophisticated estimates of chlorpyrifos loading. These aggregate statistics are therefore referred to as “local use” and “potential exposure” rather than “exposure.” Actual exposures are likely to be some fraction of these estimated potential exposures. However, the regression models used below are still informative concerning distributive EJ. For example, if the models indicate that group *A* communities have on average 50% greater potential exposure to chlorpyrifos than contrast group *B*, this supports (although not conclusively) a prediction that group *A* communities have 50% greater *actual* exposure than group *B* communities. This might further indicate, for instance, that absolute exposure models designed around members of *B* seriously underestimate exposures for members of *A*.

Mackay et al. [50] estimate the CTD for chlorpyrifos to be 62 km. For robustness, this study considered five CTD values, ranging from 1 km to 90 km. Separate models were constructed for each of the five CTD values listed in Table 2, as well as for tracts and places. These two methodological choices give 5×2=10 models. See also Table 3 and Figure 2.

### 2.5. Effects Estimation

This study used an effects estimation approach, rather than an hypothesis testing approach [51]. Specifically, the regression specifications discussed below are designed to avoid or mitigate bias due to spatial correlation, and a bootstrap method is used to account for errors in the independent variable measures. In addition, two major researcher degrees of freedom [52]—geographic unit of analysis (tracts vs. places) and choice of CTD value—are tracked by analyzing 10 combinations of geography and CTD value in parallel.

In this effects estimation paradigm, interval estimates of parameter values are valuable because they simultaneously report both parameter estimates and the uncertainty or precision of those estimates. Interval estimates are not interpreted dichotomously, i.e., the interpretive question is not whether the interval contains 0, because this question is equivalent to testing a null hypothesis [51]. Rather, the interpretive question is what the collection of interval estimates from several regression models indicate about the compatibility of a range of values with the data and modeling assumptions [53]. Robustness reasoning is especially important in this paradigm: if a set of models all agree on an estimate, this increases our confidence in the estimate [54].

### 2.6. Independent Variable Selection

Non-spatial exploratory data analysis indicated that, for almost all (>80%) tracts and places, almost all residents (>80%) were either Hispanic or non-Hispanic White. Thus, Hispanic and non-Hispanic White proportions are strongly negatively correlated (r=−0.9), and so non-Hispanic White proportion was dropped from the independent variable list.

Four control variables were used. Because PUR data come only from agricultural uses, we expect chlorpyrifos use to be negatively correlated with population density and positively correlated with agricultural employment. Because density is left-bounded at 0 and ranges over several orders of magnitude, log density was used. To account for residual spatial heterogeneity, discussed below, county-level population density and agricultural employment variables were also included as controls.

All independent variables are given in Table 4, and descriptive statistics for both places and tracts are given in Table 5.

### 2.7. Regression Specification

Exploratory data analysis indicated that local chlorpyrifos use values are non-Gaussian and range over multiple orders of magnitude across the entire study area. In addition, for lower CTD values (1, 10), the left bound at 0 creates highly asymmetrical distributions. A log transformation of the DV appeared to mitigate the left bound for the lower CTD values without qualitatively changing the distributions for the higher CTD values. Base 10 was used for interpretability, so that dependent variable units are locally weighted pound orders of magnitude.

A comparative analysis of three regression specifications indicated that a spatial Durbin model, which incorporates spatial lags for both dependent and independent variables, was appropriate for these data [55]. Further discussion of this model, the comparative analysis, and the selection of 3-nearest-neighbor spatial weights, is given in the electronic supplement.

### 2.8. Measurement Error and Bootstrap

ACS estimates can have large margins of error, especially for subpopulations of difficult-to-survey rural tracts [56]. This kind of measurement error can induce *attenuation bias* or *regression dilution*, in which correlation estimates are biased towards 0 [57]. In the context of dEJ analysis, attenuation bias is potentially a serious problem, insofar as it leads to the underestimation of environmental disparities. That is, measurement error can make distributive environmental injustices seem less severe than they actually are.

A bootstrap approach was developed to account for the effect of measurement error on the parameter estimates. In a “basic” or unparameterized bootstrap, samples are taken from the observations in the dataset (with replacement), forming a resampled dataset of the same size as the dataset [58]. These samples approximate drawing a new sample from the original population. By calculating estimates on a set of resampled datasets (1000 resamples is common), we can estimate sampling distributions of population statistics. For example, if ${\hat{\beta }}^{1},{\hat{\beta }}^{2},\ldots ,{\hat{\beta }}^{1000}$ are regression coefficient estimates calculated on 1000 resampled datasets, ${\mathrm{sd}}_{l}\left({\hat{\beta }}^{l}\right)$ gives an estimate of the standard error of the coefficient estimate and quantiles of the ${\hat{\beta }}^{l}$ distribution give estimated quantiles of the parameter’s sampling distribution.

In the present study, the uncertainty of concern is with the ACS estimates for the IV values. Assuming normality for these estimates, “resamples” can be drawn from the Gaussian distribution centered at the ACS point estimate with standard deviation the ACS margin of error. Note that this approach resamples “at” rather than “from” locations. The set of locations, with their spatial relations, is treated as fixed (the given sets of tracts and places), and at each location a new “observation” of the IV values is simulated. By contrast, the unparameterized bootstrap assumes a hypothetical infinite population of potential locations, and simulates drawing a new set of locations from this population. In a spatial context, the unparameterized bootstrap requires careful handling to account for spatial dependence [59].

Five hundred resampled datasets were constructed for each combination of geography (places or tracts) and CTD value (total 500×2×5=5000 resampled datasets). For each resampled dataset, a spatial Durbin model was fit and the following statistics were calculated: impacts for all independent variables; ρ, the coefficient on the lagged dependent variable; and Moran’s *I* on the model residuals. Impacts (see below) were calculated using a Monte Carlo method, with 100 draws for each resampled dataset. These Monte Carlo draws were then combined at the geography–CTD level, producing 100 × 500 = 50,000 draws for the impacts for each IV–geography–CTD combination. Medians and quantiles for these resampled distributions are reported below.

## 3. Results

Except when noted, the analysis in this section focuses on the resampled dataset and its models.

Impacts are reported, rather than coefficients, to account for spatial feedback in the effects of independent variables (§2.7, [55]). For further discussion, see the electronic supplement. Impacts were calculated using the impacts() function in the R package spdep, which uses a Monte Carlo method based on the traces of the powers of the spatial weights matrix *W*. For the resample dataset, these Monte Carlo draws were aggregated across the resamples to produce bootstrap distributions of impact.

Figure 3, Figure 4 and Figure 5 show total impact Monte Carlo estimates for IVs for CTD values of 1, 10, and 30-60-90 respectively. Total impact estimates were made for the spatial Durbin models fit on the observed dataset as well as each bootstrap resample of the IV estimates. Impact estimates for the resampled datasets were combined into a single estimated sampling distribution for each geography–CTD–IV combination.

Estimates are generally much more uncertain (wider percentile intervals) for smaller CTD values. Indeed, for CTD = 1 km, several estimates are uncertain across tens of orders of magnitude. By contrast, for CTD values of 30 or greater, uncertainty is often less than about 2 orders of magnitude, and in some cases substantially less than 1 order of magnitude. In short, whatever physical validity different CTD values have, higher CTD values have more precise effects estimates. The largest uncertainties are for the smallest demographic groups in the study area, namely, Asian, Black, Indigenous, and children proportions.

Figure 3, Figure 4 and Figure 5 show both the bootstrap resamples (triangles and dashed lines) and observed data/ACS point estimates (circles and solid lines). There is general agreement between both resamples and observed data for both effect point estimates and effect uncertainty. Even when the point estimates are somewhat different, the degree of uncertainty (width of the 5–95% interval) is often similar.

The remainder of this discussion focuses on total impact estimates for Hispanic, poverty, and agricultural employment proportion, and population density, across CTD values of 30-60-90 km, using the bootstrapped estimates to account for measurement error. These estimates are reported in Figure 6 and (for CTD 60 km only) Table 6. Transformed estimates are reported to aid interpretation: when the estimated total impact ζ is transformed as $\zeta_{trans}=10^{\zeta /10}$, $\zeta_{trans}$ can be interpreted as the multiplicative change in local chlorpyrifos use associated with a 10% increase in the corresponding IV. For example, if $\zeta_{trans}=1.5$ for some proportion variable, then a 10-percentage-point increase in this proportion is associated with a 50% increase in local chlorpyrifos use.

Impact estimates for agricultural employment and poverty proportions differ across different types of geography. For agricultural employment, for tracts, all intervals are entirely or almost entirely above 1; while for places all intervals are centered near 1, and so are compatible with positive, negative, and negligible or null effects. For poverty, 4 of 6 intervals are centered near 1; 2 intervals, both for places, are centered around 1.05, but extend below 1.

In contrast, estimates for population density agree across geography types, with interval endpoints ranging from as low as 1.01 to as high as 1.16. Estimates for tracts are more precise than those for places, giving a narrower range of 1.04–1.13-fold. Across both geography types, estimates are closer to 1 at greater CTD values.

Finally, estimates for Hispanic proportion generally agree across geography types. A positive association appears across all CTD values, in both observed and resampled datasets, and in 6 out of 11 county-level models (Fresno, Kern, Solano, and Tulare counties). For tracts, the estimates of these effects range from 1.05 to nearly 1.4.

## 4. Discussion

### 4.1. Discussion of Selected IVs

For agricultural employment and poverty, estimates were compatible with positive, negative, and negligible effects, with differences in trends across geography types. Disagreement in estimates between tracts and places may be due to the way places were constructed, namely, as a way to capture relatively dense population centers. This process might have excluded many agricultural workers and the rural poor, which in turn might lead to biased effect estimates; though Table 5 indicates that mean agricultural employment and poverty proportion were *greater* in places than in tracts. County-level heterogeneity may also have been a factor. For example, poverty appears to have had a positive effect for places in Stanislaus and Tulare counties, a negative effect in Butte county, and perhaps a negative effect in Kern county (Appendix A).

The positive effect of population density is counterintuitive: since chlorpyrifos is used primarily in agricultural areas, with low population density, we would expect to see a negative association. Local knowledge of the Central Valley suggests one potential explanation. As indicated by Figure 2, with a CTD of 60, chlorpyrifos use is consistently much greater in the San Joaquin Valley, the southern part of the Central Valley. The San Joaquin Valley has several small- and medium-sized cities, including Fresno (population approximately 500,000), Bakersfield (400,000), Stockton (300,000), and Modesto (200,000). By contrast, after excluding Sacramento, the largest cities in the Sacramento Valley (the northern part of the Central Valley) are Redding and Chico (both approximately 90,000). Population density may therefore be confounded, at least in part, with large-scale county or regional differences in chlorpyrifos use due to the kinds of crops grown or the prevalence of insect pests.

However, the positive effect of population density remains apparent in most of the county-level regressions (Appendix A). In Butte and Solano counties (Sacramento Valley) as well as Fresno, Kern, and Tulare counties (San Joaquin Valley), the regression models indicate that potential chlorpyrifos exposure increases with population density. This effect is ambiguous for Stanislaus county (San Joaquin Valley), where there appears to be a larger but relatively highly uncertain effect for places but no clear effect for tracts. And population density is negatively associated with potential chlorpyrifos exposure in San Joaquin county (San Joaquin Valley).

Another potential explanation is that growers with better access to capital may use pesticides and other inputs more intensively. Fields and orchards near urban areas are likely to have higher land values, and more valuable land is more likely to be owned and managed by wealthier growers. Public land ownership and appraisal data might be used to examine these impacts of land value and ownership. However, growers are anonymized in the DPR data, and uses are coarse-grained to 1 mile-square (1.6 km × 1.6 km) sections. Such analysis therefore falls outside of the scope of the current study.

Within the scope of this study, the evidence of a positive association between Hispanic proportion and potential chlorpyrifos exposure was robust, with agreement across choices of parameter values, model specifications, and geographic units of analysis. Using estimates for tracts for CTD of 60 km (1.07–1.21), a 60-point difference in Hispanic proportion—corresponding to the difference between a Hispanic-minority and Hispanic-majority tract—would be associated with a 1.5–3.1-fold increase in potential chlorpyrifos exposure.

As noted in the introduction, that literature review did not find any prior studies that applied spatial methods to analyze environmental justice aspects of pesticide use or potential pesticide exposure. We are therefore unable to compare these results with those of other work.

### 4.2. Context of Chlorpyrifos Use

Chlorpyrifos is not the only pesticide used in large quantities in California during the study period. In 2015, chlorpyrifos was the most heavily used acetylcholinesterase inhibitor (ACI) in the state (1.1 million pounds active ingredient); but 10 other ACIs were used in large quantities (more than 100,000 lbs active ingredient), including acephate, bensulide, carbaryl, dimethoate, EPTC (S-Ethyl dipropylthiocarbamate), ethephon, malathion, methomyl, naled, thiobencarb. A total of 4.5 million pounds of active ingredient ACIs were used in California in 2015 (Table 7, California Department of Pesticide Regulation [60]).

As noted above, during the study period chlorpyrifos was most heavily used on almonds: on average, 353 thousand pounds active ingredient were applied per year during 2013–2015 (Table 4, [47]), on approximately 190 thousand acres per year at a rate of 1.8 pounds active ingredient per acre (Table 5, [47]). In terms of acreage, other insecticides as well as herbicides were used more heavily on almonds in California in 2015, including glyphosate, oil-based insecticides, and abamectin (more than 1 million acres treated for each) (Figure 16, California Department of Pesticide Regulation [60]).

Chlorpyrifos was used throughout the year in 2013–2015 in California, though usage was typically higher in May–August (on average 54% of total use by pounds active ingredient) and lower in October–January (15%) (Table 7, [47]), loosely corresponding to the almond season in the Central Valley.

### 4.3. Limitations

Four limitations of this study are worth noting. First, there are limitations in the DPR public use data. This dataset covers agricultural uses only, and does not include industrial, commercial, state, or residential use of pesticides. This may not be an issue for chlorpyrifos, which is banned for residential use in the US and is used almost exclusively for agricultural purposes (Table 2, Segawa and Wofford [47]). However, for other pesticides that are widely used in sectors not covered by the DPR data, the DPR data are likely to have significant gaps. For example, glyphosate is widely used in residential settings, and these uses would not appear in the DPR data. In addition, the DPR dataset tracks active ingredients, not the complex mixtures of product formulations that may enhance or mitigate active ingredient toxicity.

Second, the methods used here model local use as a proxy for potential chlorpyrifos exposure, not actual exposure, and effectively assume a highly simplified fate-and-transport model. More sophisticated fate-and-transport models may help remove this unmodeled uncertainty.

Third, the spatial distribution of population data does not model occupational exposure, children at school, or “take-home” occupational exposure (e.g., an agricultural worker brings home contaminated clothes that are handled by her children) [61].

Fourth, there are signs of heteroscedasticity and residual spatial autocorrelation in the spatial Durbin models, especially for places. There were indications throughout the study that county-level effects would address the non-Gaussian patterns in the data. Other data sources might be incorporated to account for background baseline chlorpyrifos use rates, such as nearby crop species cultivated. Or spatial random effects models—which allow effects to vary across space—might be used.

### 4.4. Potential Policy Implications

This study found consistent evidence that a 10-point increase in Hispanic population proportion was associated with a 1.05–1.4-fold increase in local chlorpyrifos use. Using these estimates, a 60-point difference in Hispanic proportion would be associated with as much as a 6-fold increase in potential chlorpyrifos exposure.

Reflecting on these results together with the long-term neurotoxic effects of chlorpyrifos suggests the possibility that Hispanic communities in the Central Valley may be subject to a process of cumulative disadvantage [62]. Pannu [63] argues that ethnicized local political processes in the Central Valley have led to the marginalization of Hispanic residents in local water politics. Because California’s pesticide regulatory system delegates significant power to county-level agricultural commissioners (p. 13, [64]), it is highly plausible that the same ethnicized political dynamic is also present in local pesticide politics.

Consider the following scenario. (1) Ethnic residential segregation leads to differential chlorpyrifos exposure in Hispanic communities, whether through the processes described in the previous paragraph or otherwise. (2) Chlorpyrifos exposure impairs the cognitive abilities of the children of these communities, compared to their peers in non-Hispanic White communities. (3) These cognitive impairments reduce merit-based educational opportunities for Hispanic children as they grow up, which in turn (4) reduces the social capital (e.g., number of bachelor’s degree holders) available to Hispanic communities. Finally, closing the loop, (5) because of this reduced social capital, the concerns of these communities are dismissed as anecdotes or misinformed [24], making them more politically vulnerable—and thereby exacerbating ethnic differences in chlorpyrifos exposure (1).

While empirically validating this complete scenario is beyond the scope of the present study, each step has at least some empirical support. The present study provides support for step 1. Step 2 is supported by the available research on the neurotoxicological effects of chlorpyrifos. Steps 3 and 4 are at least highly plausible. Regarding step 5, analyzing the discourse used in efforts to regulate methyl iodide and chloropicrin, Guthman and Brown observe that consumer-oriented rhetorical frames appear to be more politically effective than worker-oriented rhetorical frames [65,66]. These frames have strong class-ethnic associations with white collar non-Hispanic White consumers and Hispanic farmworkers respectively [67]. Despite the disadvantages suggested by this scenario, Hispanic-led environmental justice organizations have had some notable successes in California’s pesticide politics (pp. 216–221, [68,69,70]). But by placing the organizational burden on the communities most likely to be harmed by the pesticide, the environmental injustices inflicted by the spatial distribution of chlorpyrifos are magnified rather than mitigated.

The plausibility of this scenario indicates a need for social science methods and expertise in pesticide risk assessment. Steps 3–5 describe social and political processes that cannot be reduced to biophysiological processes operating within individual organisms. Methods from experimental biological sciences are appropriate for certain steps of the scenario; as are methods developed in epidemiology and public health. But assessing steps 3–5 requires social science methods and expertise that are not widely used in human health risk assessment [71].

## 5. Conclusions

By applying spatial regression methods to two administrative data sets, this study found that Hispanic communities in California’s Central Valley are associated with higher local chlorpyrifos use, and so higher potential chlorpyrifos exposure. This distributive environmental injustice may be a key stage in a cumulative disadvantage process, in which ethnic disparities in chlorpyrifos exposure exacerbate other social and economic disparities, and ultimately increase disparities in pesticide exposure even further.

## Figures and Tables

**Figure 1 ijerph-17-02593-f001:**
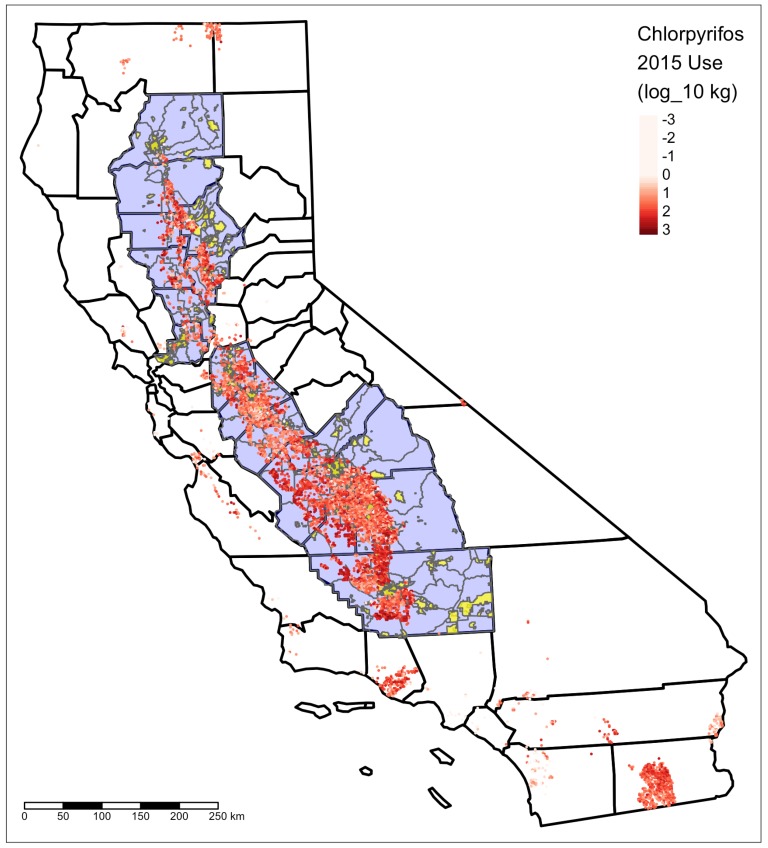
Data used in this study. Red points are chlorpyrifos use totals, shown on a log (base 10) pounds scale and for this map 2015 only. Blue regions are Census tracts included in the study area; yellow regions are included places. All California counties are shown for context.

**Figure 2 ijerph-17-02593-f002:**
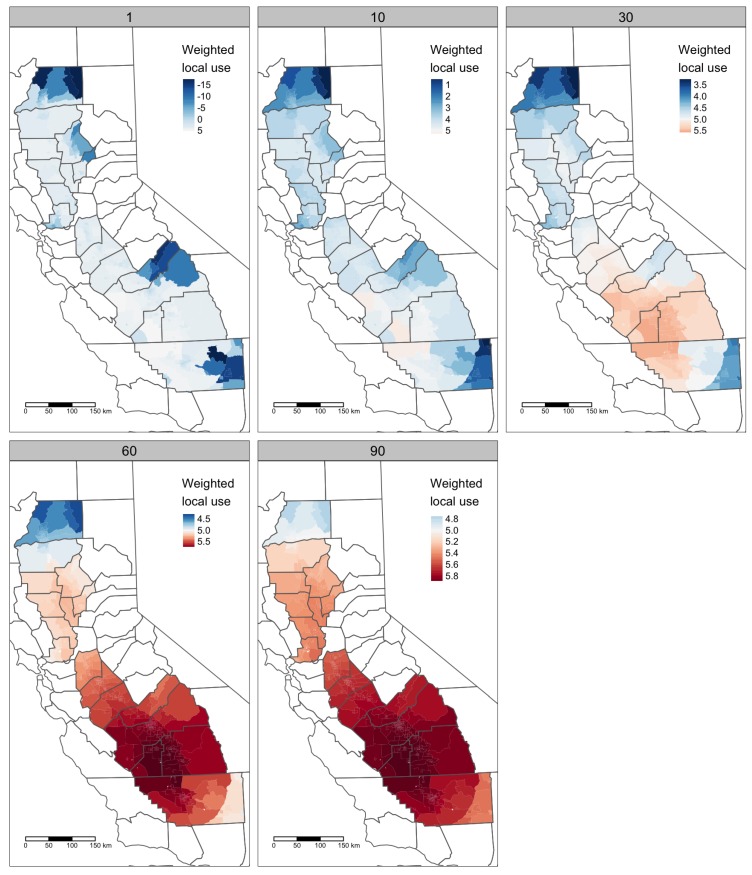
Impact of Characteristic Travel Distance (CTD) value on weighted local use values in Census tracts. Panels correspond to the different CTD values used in this study. Weighted local use is the log (base 10) of aggregate chlorpyrifos use around each tract, scaled using the decay coefficient, 2011–2015. Color scales are only roughly consistent between panels, with the scale midpoint set at 5 ($10^{5}$ = 100,000 lbs).

**Figure 3 ijerph-17-02593-f003:**
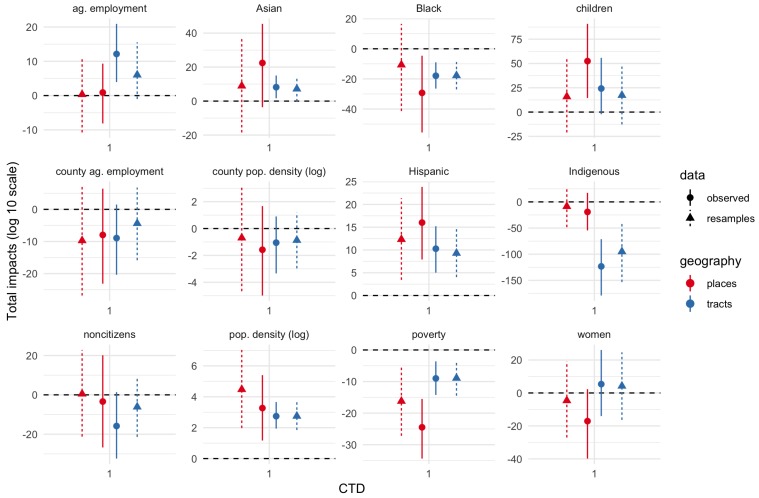
Total IV impacts, CTD = 1 km. Solid lines and circles show estimates inferred from observed data/American Community Survey (ACS) point estimates. Dashed lines and triangles show estimates inferred from bootstrap resamples to account for ACS margins of error. Tract estimates in blue; place estimates in red. Ends of line ranges indicate 5th and 95th percentiles of Monte Carlo impact draws; circles/triangles indicate medians.

**Figure 4 ijerph-17-02593-f004:**
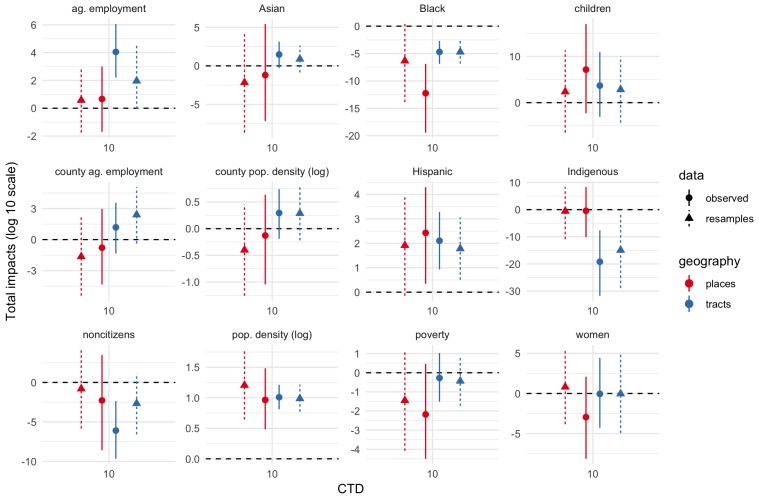
Total IV impacts, CTD = 10 km. Solid lines and circles show estimates inferred from observed data/ACS point estimates. Dashed lines and triangles show estimates inferred from bootstrap resamples to account for ACS margins of error. Tract estimates in blue; place estimates in red. Ends of line ranges indicate 5th and 95th percentiles of Monte Carlo impact draws; circles/triangles indicate medians.

**Figure 5 ijerph-17-02593-f005:**
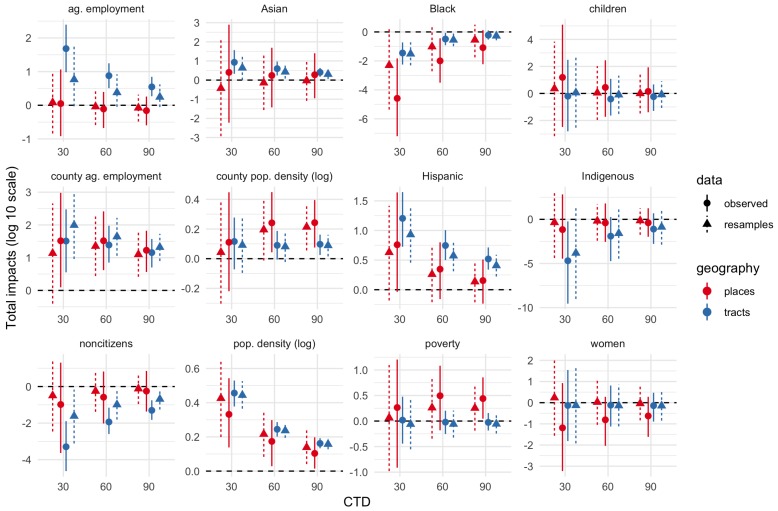
Total IV impacts, CTD = 30, 60, and 90 km. Solid lines and circles show estimates inferred from observed data/ACS point estimates. Dashed lines and triangles show estimates inferred from bootstrap resamples to account for ACS margins of error. Tract estimates in blue; place estimates in red. Ends of line ranges indicate 5th and 95th percentiles of Monte Carlo impact draws; circles/triangles indicate medians.

**Figure 6 ijerph-17-02593-f006:**
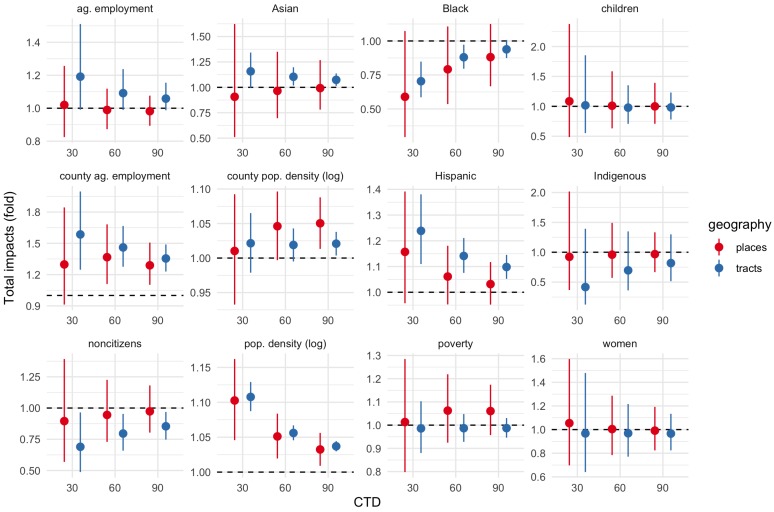
Total impact estimates from the spatial Durbin models. Total impact estimates ζ are transformed as $\zeta_{trans}=10^{\zeta /10}$. $\zeta_{trans}$ can be interpreted as the multiplicative change in local use when the corresponding IV increases by 10 percentage points. For example, if $\zeta_{trans}=1.5$, then local use is 50% greater when the corresponding IV is 10 points greater. Transformed values >1 therefore correspond to increases; values <1 correspond to decreases. Line ranges give transformed 5–95 percentile intervals of Monte Carlo impact draws. All estimates are based on the resampled datasets.

**Table 1 ijerph-17-02593-t001:** California counties comprising the Central Valley for the purposes of this study. Counties are listed roughly in north–south order. Left: counties in the Sacramento Valley (northern half of the Central Valley). Right: counties in the San Joaquin Valley (southern half).

Sacramento Valley	San Joaquin Valley
Shasta	San Joaquin
Tehama	Stanislaus
Glenn	Merced
Butte	Madera
Colusa	Fresno
Yuba	Kings
Sutter	Tulare
Yolo	Kern
Solano	

**Table 2 ijerph-17-02593-t002:** Characteristic Travel Distance (CTD) values used in this study, and corresponding decay-rate values β.

CTD (km)	β
1	0.370
10	0.905
30	0.967
60	0.984
90	0.989

**Table 3 ijerph-17-02593-t003:** Summary statistics for weighted local use values, by characteristic travel distance (CTD) and geography. Mean, standard deviation, minimum, and maximum in logged pounds.

CTD	Geography	Mean	sd	Min	Max	Moran’s *I*
1	places	−0.19	5.42	−22.69	4.47	0.88
10	places	4.26	0.94	0.41	5.21	0.94
30	places	5.30	0.46	3.53	5.87	0.97
60	places	5.75	0.35	4.61	6.20	0.99
90	places	5.98	0.30	5.07	6.34	0.99
1	tracts	1.38	2.88	−17.62	5.57	0.74
10	tracts	4.43	0.71	0.41	5.60	0.95
30	tracts	5.34	0.41	3.57	5.91	0.98
60	tracts	5.79	0.32	4.63	6.20	0.99
90	tracts	6.01	0.27	5.08	6.34	1.00

**Table 4 ijerph-17-02593-t004:** Independent variables (IVs) used in this study. All race–ethnicity groups other than Hispanic are non-Hispanic. All IVs other than population density are proportion of total population in the tract or place.

Category	Independent Variable
Race–Ethnicity	Hispanic
	Black
	Indigenous
	Asian
Children	Children under 5
Class	Income/Poverty Ratio < 1.0
Controls	log Population Density
	Employed in Agriculture

**Table 5 ijerph-17-02593-t005:** Descriptive statistics for independent variables used in this study. All race–ethnicity groups other than Hispanic are non-Hispanic. All IVs other than population density are proportion of total population in the tract or place. Moran’s *I* is calculated using 3-nearest-neighbor spatial weights.

Geography	Variable	Mean	sd	Min	Max	Moran’s *I*
places	ag. employment	0.17	0.19	0.00	1.00	0.54
places	Asian	0.03	0.05	0.00	0.42	0.29
places	Black	0.02	0.05	0.00	0.61	0.13
places	children	0.07	0.04	0.00	0.21	0.18
places	pop. density (log)	2.40	0.73	0.14	3.64	0.52
places	Hispanic	0.43	0.32	0.00	1.00	0.74
places	Indigenous	0.02	0.04	0.00	0.45	−0.01
places	poverty	0.23	0.17	0.00	1.00	0.35
tracts	ag. employment	0.10	0.13	0.00	0.65	0.65
tracts	Asian	0.08	0.08	0.00	0.53	0.62
tracts	Black	0.05	0.06	0.00	0.46	0.68
tracts	children	0.07	0.03	0.00	0.20	0.28
tracts	pop. density (log)	2.75	0.87	−0.11	3.75	0.46
tracts	Hispanic	0.42	0.24	0.02	0.98	0.79
tracts	Indigenous	0.01	0.02	0.00	0.19	0.16
tracts	poverty	0.22	0.13	0.00	0.64	0.54

**Table 6 ijerph-17-02593-t006:** Estimates of total impact (direct + indirect) from spatial Durbin models for characteristic travel distance (CTD) = 60 km. *IV*: Independent variable. Estimates and percentiles have been transformed as $\zeta_{trans}=10^{\zeta /10}$ to aid interpretation. $\zeta_{trans}$ can be interpreted as the multiplicative change in local use when the corresponding IV increases by 10 percentage points.

IV	Geography	Estimate	95% Interval	
ag. employment	places	0.99	0.87	1.12
ag. employment	tracts	1.09	0.99	1.24
Asian	places	0.97	0.70	1.35
Asian	tracts	1.10	1.02	1.20
Black	places	0.79	0.53	1.11
Black	tracts	0.88	0.80	0.97
children	places	1.01	0.64	1.59
children	tracts	0.98	0.70	1.35
county ag. employment	places	1.37	1.11	1.68
county ag. employment	tracts	1.46	1.28	1.67
county pop. density (log)	places	1.05	1.00	1.10
county pop. density (log)	tracts	1.02	0.99	1.04
Hispanic	places	1.06	0.95	1.18
Hispanic	tracts	1.14	1.07	1.21
Indigenous	places	0.96	0.57	1.49
Indigenous	tracts	0.69	0.36	1.35
noncitizens	places	0.94	0.73	1.23
noncitizens	tracts	0.80	0.66	0.95
pop. density (log)	places	1.05	1.02	1.08
pop. density (log)	tracts	1.06	1.05	1.07
poverty	places	1.06	0.92	1.22
poverty	tracts	0.99	0.93	1.05
women	places	1.01	0.79	1.29
women	tracts	0.97	0.77	1.22

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
