# Peer review of "Census Demographics and Chlorpyrifos Use in California’s Central Valley, 2011–15: A Distributional Environmental Justice Analysis"

_ijerph, 2020, doi:10.3390/ijerph17072593_

Round 1

Reviewer 1 Report

In this manuscript, the author used spatial regression techniques to examine the distribution of chlorpyrifos use across California’s Central Valley. The topic of the manuscript is interesting especially with a widely used organophosphorus pesticide like chlorpyrifos. But, the manuscript needs some more organization and some concerns need to addressed to fit for publication as follows:
1. Abstract:
- A clear conclusion of the study should be added.
- Line 6: (DPR) and (PUR) abbreviations should be deleted as they have not been repeated again.
2. Keywords are missed.
3. Introduction:
- More details of the different sources of human exposure to chlorpyrifos and environmental distribution should be added.
- Line 23: regarding, [see also 3, 67; 4; 5; 6; and further citations in 7] the paragraph describe the study of Bellinger. Thus, what these references refer to? Please, revise and rephrase.
- Line 88: "this study" refers to the study of Liévanos or what? Please, clarify.
- If studies related to the subject in 2019 exist, please update your data.
4. Materials and methods:
- The data in Tables 1 and 2 not need to be demonstrated in a table, just mention in the text in two paragraphs.
- Table 3: mention the full term of CTD in the legend.
5. Results:
- Line 280: mention the number of figures.
- Table 6: remove the column of CTD as it has been clarified in the legend. Also, mention the full term of CTD in the legend.
6. The discussion needs to be deepening through comparison to the earlier studies and more possible interpretations of the findings of the study.
7. Conclusion: should be fortified with recommendations and further perspectives.

Reviewer 2 Report

The manuscript entitled «Census Demographics and Chlorpyrifos Use in California’s Central Valley, 2011-15: A Distributional Environmental Justice Analysis» has been reviewed.

Overall, this is an original, scientifically sound research of worldwide scientific / technical interest, using correct methodology and data analyses. The results, clearly presented, are well connected to Discussion. There are some few revisions that should be made before to accept the manuscript for publication. All required revisions are stated below:

L15-16:  where are the keywords ? please add them

L18: please replace "Chlorpyrifos is one of...... "    with   "Chlorpyrifos, an Acetylcholinesterase inhibitor in insects, is one of...... "

L23:  the reference "67" is misplaced here, please delete it

L29:  delete the second  "and"

L30:  add  "active substance"  after  "chemical"

L70: you said "a major agricultural region" . In this context, can you provide more info regarding the most important crop(s) cultivated there ? (e.g. mainly a grape-growing region ?? or what exactly ??). This info is important to know against which kind of insect pest species chlorpyrifos has been applied there

L78:  add  "human"   before   "health"

L84-85:  your statement is not sufficiently informative to the reader; please give some examples (e.g. names of the chemical active substances used) of the  "other kinds of environmental health hazards"

L101-104:  please provide further info linked to the main crops cultivated in California's Central Valley (example: grapevine, .....)

L114:  replace "17"  with  "Seventeen"

L146:  replace "9"  with  "Nine"

L185: can you further clarify here the practical difference between "potential exposure”  and  “exposure” 

L263:  replace  "500"  with  "Five hundred"

L305: DISCUSSION - general comment: I would suggest authors to add further data about:

    (i) other commonly used pesticide active substances in the study areas (in addition to chlorpyrifos)

    (ii) the most commonly used field (recommended, registered) rates of chlorpyrifos for controlling crop pests in the study area (i.e. dose of active substance per acre)

    (iii) the periods (or seasons) during which there is higher human exposure to chlorpyrifos following its application against crop pests in the study areas

L328:  which are the  "other pesticides" widely used in sectors not covered by the DPR ?? please give examples of active substances and related references

Author Response

Please see the attachment. Responses to Reviewer 2 begin on page 3.

Round 2

Reviewer 1 Report

-